# The Diagnostic Challenge of Osteoid Osteoma in the Bones of the Hand—A Case Series

**DOI:** 10.3390/diagnostics13071279

**Published:** 2023-03-28

**Authors:** Jasmin Meyer, Matthias Priemel, Tim Rolvien, Karl-Heinz Frosch, Carsten Schlickewei, Sinef Yarar-Schlickewei

**Affiliations:** Department of Trauma and Orthopedic Surgery, University Medical Center Hamburg-Eppendorf, 20251 Hamburg, Germany

**Keywords:** osteoid osteoma, bone tumor, benign bone tumor, hand tumor, tumor surgery

## Abstract

Osteoid osteoma (OO) is a benign bone tumor that rarely occurs in the bones of the hand. Due to the comparatively non-specific symptoms when occurring in the hand, OO is often misdiagnosed at first presentation, posing a diagnostic challenge. In the present case study, six cases of phalangeal and carpal OO, treated surgically at our department between 2006 and 2020, were retrospectively reviewed. We compared all cases regarding demographic data, clinical presentation, imaging findings, time to diagnosis, surgical treatment, and clinical outcome in follow-up examinations. When OO occurs in the bones of the hand, it can lead to swelling and deformities, such as enlargement of the affected bone and nail hypertrophy. Initial misdiagnoses such as primary bone tumors other than OO, tendinitis, osteomyelitis, or arthritis are common. Most of the presented cases showed a prolonged time until diagnosis, whereby the primarily performed imaging modality was often not sensitive. CT proved to be the most sensitive sectional imaging modality for diagnosing OO. With adequate surgical treatment, complications and recurrence are rare.

## 1. Introduction

Osteoid osteoma (OO) was first reported by Jaffe in 1935 [1]. It is a common benign bone tumor mostly present in long tubular bones of the lower extremities. The femur and tibia are most commonly affected, with OO often located in the cortical bone of the diaphysis. Overall, OO accounts for about 12% of all benign bone tumors [2,3]. Men are affected more frequently, while most of the patients are under 30 years of age [4,5]. The typical feature on imaging, preferably computed tomography (CT), is a central round lucent area called the nidus surrounded by an osteosclerotic rim [6,7]. The nidus contains unmineralized bone called osteoid [8].

Considering the bones of the hand, the phalanges, especially the proximal phalanges, are most often affected [9,10]. Typical symptoms are pain, night pain, and improvement due to NSAID [11]. In some cases, complementary symptoms such as swelling and deformities (nail hypertrophy, nail and bone deformity) can be observed [12,13,14]. A previous evaluation of 367 OO revealed a frequency of 8.4% in the bones of the hand [15]. It is known that patients with OO of the hand often must endure a long period of discomfort until final diagnosis [16]. Common misdiagnoses include tendinitis, arthritis, osteomyelitis, exostosis, and parosteal proliferations such as Nora’s lesion [17,18]. OO in the carpal bones often initially suspect an unclear ulnar wrist pain, TFCC (triangular fibrocartilage complex) pathologies or ganglia [19].

The aim of this case series is to highlight diagnostic difficulties and the risk of misdiagnosis in the OO of the hand. In addition, the specific clinical presentation of OO in the hand bones will be discussed.

## 2. Methods

Six patients with OO in the hand bones who underwent surgical treatment in our clinic between 2006 and 2020 were included in this case series. All patients were male, with a mean age of 27.3 years (range 22 to 35 years). All patients underwent conventional radiography (X-ray), magnetic resonance imaging (MRI), and CT preoperatively to confirm the diagnosis. Clinical examinations were carried out pre- and postoperatively on patient history and complaints such as pain, movement restrictions, swelling, and deformity. All patients were surgically treated in the Department of Trauma and Orthopedic Surgery, University Medical Center Hamburg-Eppendorf. The diagnosis of OO was histologically confirmed by the Institute of Pathology, UKE Hamburg.

The 6 cases were retrospectively reviewed. The following data were collected: (1) basic data (age, gender, handedness, occupation); (2) clinical presentation (duration of symptoms, pain, night pain, response to NSAID, swelling, deformities, movement restriction, previous operations); (3) tumor presentation in imaging (affected bone, localization, soft tissue-, bone- and periost reaction, edema); (4) treatment and follow-up (surgical procedure, postoperative complications, and outcome).

## 3. Results

### 3.1. Case 1

In June 2006, a 35-year-old male patient presented to our clinic with pain and enlargement of the proximal phalanx of the left middle finger. He was right-handed and had no motoric deficits in the adjacent joints of the affected finger. The following X-ray revealed a bony lesion adjacent to the cortical bone with a strong periosteal reaction in the proximal phalanx. An open biopsy was performed. Due to the clinical presentation and histological assessment of the biopsy, Nora’s lesion was suspected. The patient received regular clinical and radiographic monitoring. In 2007 an X-ray showed a complete regression of the lesion after the biopsy. Due to permanent pain and swelling with limitations at work, MRI (Figure 1A) and CT (Figure 2A) were performed in 2009, which revealed an intracortical nidus (diameter 9 mm) in the proximal phalanx, deformity of the affected bone and edema of the surrounding tissue in MRI. In August 2010, the lesion was removed en bloc through a mid-lateral approach. The defect was filled with a bone graft from the ipsilateral iliac crest and stabilized with three tension screws. In the following histological assessment, OO was confirmed. During follow-up examinations, the patient was free of pain with a slight limitation of flexion of 5 degrees in the proximal interphalangeal joint.

### 3.2. Case 2

A 27-year-old right-handed man complained of pain in his right wrist without trauma for 7 months. The pain intensified at night and did not improve significantly with NSAID. The clinical presentation revealed tenderness of palpation over the ulnar wrist as well as pain in full extension and flection without movement restrictions. There was no swelling, deformities, or previous operations in anamnesis. An MRI (Figure 1B) performed in January and March 2020 showed excessive marrow edema in the hamate and the surrounding soft tissue. An X-ray in February 2020 showed no abnormalities in the wrists pictured. With the diagnosis of tendinitis due to overuse, the wrist was immobilized for several weeks at a peripheral outpatient clinic. The pain increased despite immobilization, and the follow-up examination showed a demineralization of the hamate. A CT scan in June 2020 (Figure 2B) revealed an intracortical nidus in the hook of the hamate with a diameter of 6 mm. As OO was suspected, the patient was referred to our clinic for surgical treatment. We performed open curettage via incision above Guyon’s canal in November 2020 (Figure 3). The defect was filled with cancellous bone from the ipsilateral distal radius. Histological analysis confirmed the diagnosis of OO (Figure 4). During the follow-up examination, the patient showed a significant decrease in pain and no motor restrictions with complete remineralization of the hamate.

### 3.3. Case 3

In January 2020, a 22-year-old male patient was referred to our clinic for the excision of an unclear tumor in the capitate of the left hand. The right-handed patient described pain in his left wrist, which had increased in the last 1.5 years. The examination showed tenderness of palpation over the wrist with painful movement restrictions in flexion and extension without local swelling. Supination and pronation were without limitations. An MRI in August 2019 (Figure 1C) revealed a small lesion in the capitate surrounded by an excessive bone marrow edema in the capitate, the hamate, the surrounding carpal joints, and the soft tissue of the dorsal wrist, including the synovial sheath of the dorsal tendon. A CT scan performed in October 2019 (Figure 2C) showed a juxta-articular lesion with a diameter of 7 mm and regional destruction of the surrounding dorsolateral cortical bone. OO was suspected. Due to pain and movement restrictions, the indication of local excision of the tumor was provided. In January 2021, an open curettage with a subsequent filling of the defect with cancellous bone from the distal radius was performed through a dorsal approach. Due to the risk of postoperative instability of the amphiarthrosis between the capitate and hamate, a temporary arthrodesis was performed using K-wires. Histological analysis confirmed the diagnosis of a subchondral OO. After surgery, the patient was free of pain with good mobility in the wrist and a slight local swelling in a four-month follow-up examination.

### 3.4. Case 4

A 27-year-old man was referred to our clinic due to painful swelling of the lower part of the middle finger of his left hand, including the proximal interphalangeal joint, for a 12-month duration. The swelling led to movement restrictions in the joints of the affected finger. An MRI in March 2019 (Figure 1D) revealed a bone marrow edema in the distal phalanx and the surrounding palmar soft tissue. An orthopedist initially ruled out a rheumatic disease. Subsequently, the patient was treated antibiotically due to the detection of antibodies against the borrelia antigen. Due to permanent pain and swelling, a CT scan (Figure 2D) was performed, which showed juxta-articular osteolysis with surrounding sclerosis. A biopsy performed in the same month showed no evidence of bacterial inflammation, and no histological diagnosis could be secured. We opted for a complete resection of the tumor with open curettage via palmar access. Histological analysis confirmed the diagnosis of an OO. The patient was free of pain without movement restrictions in the follow-up examination.

### 3.5. Case 5

A 24-year-old male patient reported enlargement and pain in the distal phalanx of the right thump for more than one year. The pain was dominant at nighttime and responded to NSAID. The clinical examination revealed local tenderness on palpation, no movement restrictions, and a deformity of the right thump with hypertrophy of the nail. Circular osteolysis at the base of the distal phalanx was initially detected in the X-ray. Osteitis was suspected, and the patient received oral antibiotics for 2 weeks and immobilization of the finger. Due to permanent pain, an MRI and CT were performed in November 2011 (Figure 1E and Figure 2E), which revealed a bone marrow edema and a small intracortical nidus close to the base of the distal phalanx. OO was suggested, and the patient introduced himself for the local resection in our clinic. We performed an open curettage via a mid-lateral incision. The patient was free of pain in follow-up examinations.

### 3.6. Case 6

A 29-year-old left-handed male patient was admitted to our outpatient clinic with a 12-month history of pain responding to ibuprofen and swelling of the proximal phalanx of the left thumb. There were movement restrictions in the metacarpophalangeal and interphalangeal joints of the thumb. An MRI in October 2019 (Figure 1F) revealed a bone marrow edema of the proximal phalanx as well as an intracortical nidus with a reaction of the adjacent periost. The additional CT (Figure 2F) confirmed the nidus, measuring about 9 mm in size, surrounded by dense reactive bone with enlargement of the middle phalanx. A primary bone tumor (differential diagnosis osteomyelitis) was suspected. An open curettage through a mid-lateral dorsal approach with removal of sclerosis revealed brown cancellous bone. OO was histologically confirmed. The patient has remained symptom-free at the follow-up evaluation.

### 3.7. Summary of the Patient Cohort

The tumor occurred in the proximal phalanges in three of the reported cases (cases 1, 4, and 6). There was one case in each of the hamate (case 2), capitate (case 3), and distal phalanx (case 5). In two of the described cases, the tumor was found juxta-articular, and the other four OO were localized intracortical. The patient in case 1 received previous operations without histological confirmation of the tumor. He was treated with en bloc resection of the tumor. A bone graft from the ipsilateral iliac crest was used to stabilize the defect. The other five cases received surgical treatment with open curettage. In two cases, the defect was filled with trabecular bone from the distal radius. All patients were free of pain during follow-up examinations. There were no recurrences or other major complications. A summary of the 6 cases is presented in Table 1.

## 4. Discussion

The aim of this case series was to clarify difficulties and peculiarities in the diagnosis of OO in hand surgery. The bones of the hand are a rare site for OO, which is commonly diagnosed in long tubular bones such as the tibia and femur [20]. OO can be classified according to localization (i.e., intracortical, cancellous, subperiosteal, juxta-articular), with intracortical localization being the most common [20,21]. When occurring in the hand, the phalanges, especially the proximal phalanges, are most often affected [10]. In our case series, there were three cases of OO in the proximal phalanges and one each in the distal phalanx, hamate, and capitate. All patients were male, with an average age of 27.3 (range 22–37). In the available literature, men are more frequently affected, with a ratio of 2:1 and an increased occurrence in the second and third decades of life [15,22].

In the clinical presentation of the tumor, the triad of pain, nocturnal pain, and improvement due to NSAID (especially ASS) is often described. The reduction of pain by NSAID treatment can be explained by the production of prostaglandin inside the nidus [22]. All cases included in our study described the pain as the leading symptom. In contrast, nocturnal pain and improvement due to NSAID were reported less frequently. Nevertheless, the peculiarities of OO in the hand should be emphasized. Due to the small size of the bones with little soft tissue coverage, local swelling of the digit is a common symptom of OO in the phalanges [14]. Importantly, all the presented cases of phalangeal OO not only suffered from swelling of the surrounding soft tissue but additional enlargement and deformity of the affected bone were also found in all four of them.

Nail hypertrophy or deformity, as in case 5, is a common symptom of OO in the distal phalanges, and therefore further diagnostics should be considered in these cases [11,12]. Recent studies describe pronounced bone marrow edema in MRI, particularly in intra-articular OO [8]. In two of our included cases, the OO was located juxta-articular, with an excessive bone marrow edema and edema of the surrounding soft tissue, and an effusion of the adjected joints was visible in MRI. The affected patients suffered from movement restrictions in the adjacent joints. One further case of intracortical OO showed restricted mobility due to inflammatory edema in the adjacent flexor tendons. In summary, a juxta-articular location of the tumor increases the risk of restricted mobility in the adjected joint, which is relevant in the hand due to the small size of the bones and the frequent joint formation of the cartilage-coated bone interfaces.

In addition to the mentioned clinical symptoms, the diagnosis of OO is based on imaging findings. Typical findings are a central nidus with a diameter of up to 2 cm surrounded by sclerotic bone [23]. A delay in the appearance of the nidus in imaging is often and makes the diagnosis more difficult [24,25]. The primary X-ray taken after the onset of the symptoms is often unspecific or without pathological findings, as in four of the six presented cases [11]. Additional imaging should be performed when OO is suspected, with a CT scan showing the highest sensitivity in detecting OO [5,26]. The MRI is a frequently used imaging technique for unspecific hand pain. In addition to the typical nidus, bone marrow edema and edema of the adjected soft tissue are common findings in MRI imaging. An extensive bone marrow edema can mask the nidus and lead to delayed diagnosis or misdiagnosis, such as osteomyelitis or arthritis, which makes the MRI less sensitive [8]. Considering the presented cases of OO, primary MRI showed a sensitivity of 66.7%, which is lower than reported in the literature [5,27]. If findings in MRI are unspecific, an additional CT scan is recommended [28].

In the presented cases, the time to diagnosis averaged 20.5 months (range 7–62 months). In OO in the bones of the hand, misdiagnoses such as osteomyelitis and arthritis are frequently described and must, therefore, be clearly differentiated [17,18]. Looking at the cases included in this case series, some misdiagnoses should be considered further. In particular, other primary bone tumors are a possible differential diagnosis to OO (i.e., osteochondroma, enchondroma, Nora’s lesion) [9]. Nora’s lesion, also known as “bizarre parosteal osteochondromatous proliferation” (BPOP), was initially suspected in case 1 due to clinical presentation and characteristic imaging findings. However, in contrast to the OO, Nora’s lesion usually encounters the surfaces of small tubular bones of the hand and feet with an often irregular surface and the absence of continuity for the lesion with the medullary cavity of the host bone [29]. Histologically, it can be distinguished from the OO by consisting of cartilage, bone, and fibrous tissue. It has a high rate of recurrences in the case of incomplete resection [30,31]. Especially in OO in close proximity to the wrist, tendinitis, as initially suspected in case 2, is a common misdiagnosis. Tendinitis due to overuse is common in young people, and we could confirm in our cases that the edema surrounding the OO often affects the enclosing soft tissues and tendons, which can result in similar ailments as tendinitis with pain, tenderness, and movement restrictions [24]. Nevertheless, the correct diagnosis can be achieved by displaying the nidus using CT. Arthritis is also one of the frequent differential diagnoses of OO, especially when the tumor is located juxta-articular, as in cases 3 and 4. Frequent symptoms of OO, such as swelling, pain, and movement restrictions of the adjected joint, may suggest an inflammatory disease of the hand and finger joints. Although the rheumatic disease is a possible differential diagnosis, as suspected in case 5, again, visualization of the nidus on cross-sectional imaging is important in distinguishing OO from inflammatory disease. Finally, osteomyelitis should be mentioned as a common misdiagnosis, which may have similar characteristics to an OO in imaging. Especially a subacute osteomyelitis (i.e., Brodie abscess) may be difficult to distinguish showing pain, tenderness, swelling and sometimes fever as leading symptoms [32,33]. As confirmed in cases 5 and 6, an inflammatory disease should be excluded by a microbiological examination of the removed lesion.

Regarding the therapy of OO in the hand bones, surgical therapy with open curettage or en bloc resection predominates in the literature [23]. These allow the safe removal of the nidus and surrounding sclerosis. Nevertheless, percutaneous procedures such as radiofrequency ablation (RFA) show successful results in treating OO in the bones of the hand and have the advantage of low trauma [34,35,36]. A central CT-guided puncture of the nidus is necessary, whereby the close relationship to nerves, tendons, and vessels with the risk of unintentional damage should be considered [15,37]. The advantage of surgical removal consists of the subsequent histopathological confirmation of the tumor. Histopathologically, OO is characterized by a high proportion of non-mineralized bone matrix (osteoid), fibrous stroma, and no signs of necrosis or cellular atypia (Figure 4). When comparing the histological presentation of OO with Osteoblastoma, they are histologically indistinguishable [38]. However, the two bone-forming tumors are characterized by the size of their nidus, which in Osteoblastoma is defined as more than 2 cm in diameter [2]. Imaging often succeeds in distinguishing OO and Osteoblastoma from the most common primary malignant bone tumor, Osteosarcoma. This tumor with extremely high mortality appears as a lytic, sclerotic, or mixed lytic and sclerotic lesion, often combined with a reaction of the surrounding soft tissue and destruction of the cortical bone. Histologically, several subtypes can be distinguished [38]. However, recent studies have shown that in case of uncertain differentiation between benign OO or Osteoblastoma and Osteosarcoma, the diagnosis can benefit from using fluorescence in situ hybridization (FISH) analysis for FOS gene arrangement [39]. FOS, and at a lower frequency, FOSB rearrangements were found in the majority of OO and Osteoblastomas, whereas they have only been detected less frequently in osteosarcomas [38,39,40]. This provides an additional tool in the histopathological examination for differentiation of the tumor after surgical resection.

In the literature, both percutaneous and surgical therapies for OO show good outcomes with low complication and recurrence rates [11]. Although various surgical treatments were performed in our cases (en bloc resection, open curettage with or without cancellous bone graft), postoperative X-ray images show a satisfactory osseus regeneration of the defect with good clinical outcomes in follow-up examinations. However, a recommendation regarding surgical therapy requires a larger study population, including a review of the literature in order to compare the clinical outcome and recurrence rate according to surgical treatment options like en bloc resection, open curettage, and cancellous bone graft.

In conclusion, we have demonstrated that patients with an osteoid osteoma in the bones of the hand often have complaints over a long period of time until the correct diagnosis is made. The reason for this is likely the range of non-specific symptoms, which can lead to misdiagnoses like arthritis, tendinitis, or osteomyelitis. Although pain is the leading symptom, local swelling, deficits in the mobility of the adjected joints, and deformities such as nail hypertrophy and bone deformity can be observed in patients with OO, especially in the phalanges. If an OO is suspected, CT scans show the highest sensitivity when compared to other imaging modalities. In case of an unspecific bony process, we recommend regular follow-up with appropriate cross-sectional imaging since the typical nidus formation of the OO often occurs after the onset of symptoms. Despite good results in percutaneous treatment options like RFA, surgical therapy is the most common treatment option concerning OO in hand surgery and enables a reliable histological determination of the tumor together with a good clinical outcome and a low rate of recurrence.

## Figures and Tables

**Figure 1 diagnostics-13-01279-f001:**
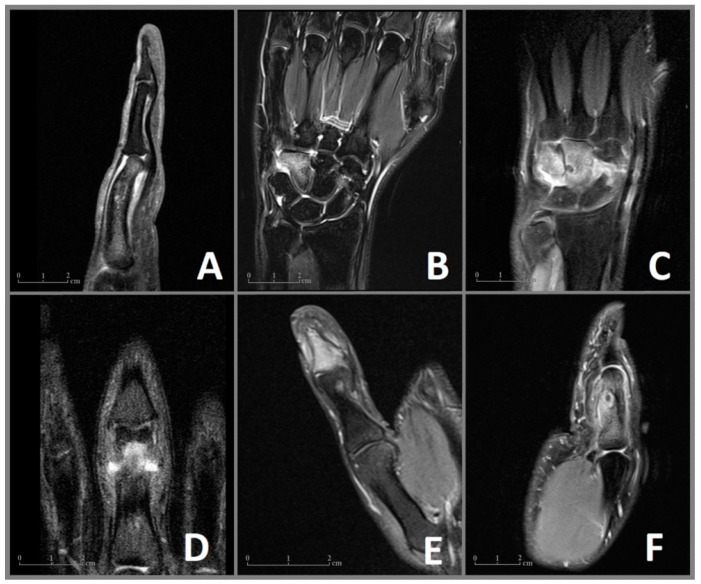
MRI imaging of the OO in cases 1–6 (**A**–**F**) with visualization of the nidus and the surrounding edema of the bone marrow and the soft tissues. (**A**) Case 1: Sagittal MRI sequence showing the nidus in the proximal phalanx with a deformity of the affected bone and edema of the soft tissue. (**B**) Case 2: Coronal MRI sequence revealing an extensive bone marrow edema of the hamate. (**C**) Case 3: Coronal MRI sequence showing the juxta-articular nidus with extensive edema of the capitate, hamate, and an effusion of the intercarpal joints. (**D**) Case 4: Coronal MRI sequence showing the nidus in the proximal phalanx. (**E**) Case 5: Coronal MRI sequence revealing a juxta-articular nidus in the distal phalanx with an effusion of the distal interphalangeal joint. (**F**) Case 6: Sagittal MRI image showing the edema of the proximal phalanx, a central nidus with a reaction of the adjected periost as well as edema of the soft tissues.

**Figure 2 diagnostics-13-01279-f002:**
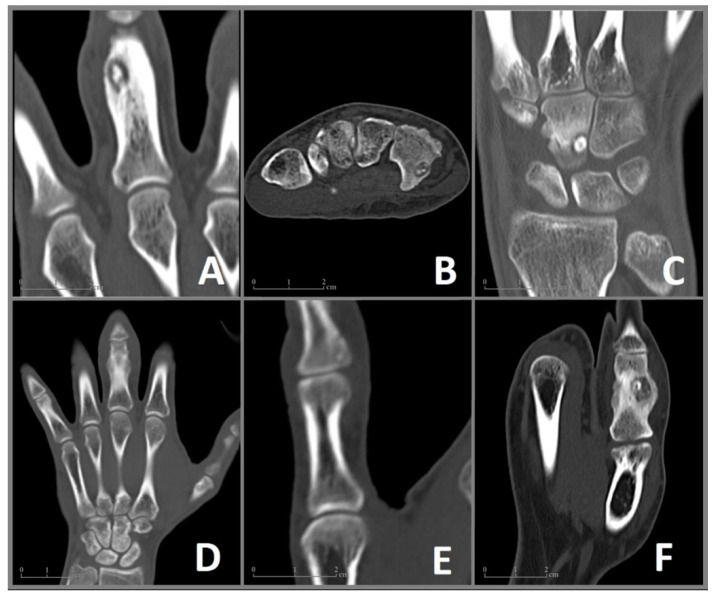
CT imaging of the included cases 1–6 (**A**–**F**) of OO in the bones of the hand. (**A**) Case 1: Coronal CT image showing a nidus with a diameter of 9 × 6 mm and surrounding sclerosis in the proximal phalanx of the left middle finger. (**B**) Case 2: Axial CT image of a 6 × 4 mm nidus in the hook of the hamate. (**C**) Case 3: Coronal CT image revealing a juxta-articular nidus (7 × 5 mm) at the ulnar surface of the capitate. (**D**) Case 4: Coronal CT image showing enlargement and deformation in the proximal phalanx of the middle finger. The nidus (10 × 7 mm) is difficult to distinguish. (**E**) Case 5: Coronal CT Image of a 6 mm nidus in the distal phalanx in close proximity to the interphalangeal joint. (**F**) Case 6: Coronal CT image showing a 9 × 8 mm nidus in the proximal phalanx of the thumb with pronounced sclerosis as well as deformation and enlargement of the affected bone.

**Figure 3 diagnostics-13-01279-f003:**
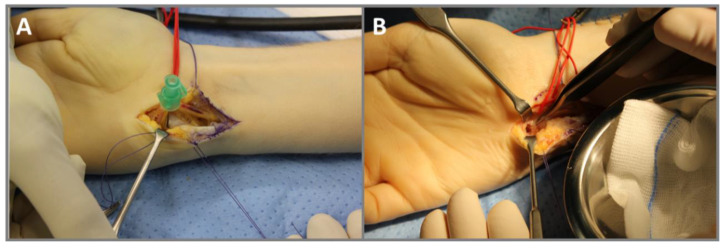
Intraoperative images of the surgical removal of an OO in the hook of the hamate in a 27-year-old male (case 2). (**A**) Approach via the Guyon’s canal. A needle is used for the position monitoring of the OO via X-ray. (**B**) Bony defect after curettage of the nidus and resection of sclerosis.

**Figure 4 diagnostics-13-01279-f004:**
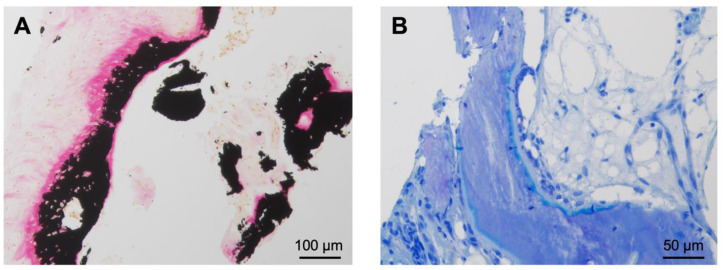
Histological findings of the resected OO from the hook of the hamate (case 2). (**A**) Von Kossa staining showing irregular bone trabeculae with high amounts of osteoid and fibrous stroma. The proportion of unmineralized bone mass (osteoid) is stained bright red. The mineralized bone next to the osteoid is stained black. No signs of necrosis or cellular atypia are seen. (**B**) In the toluidine blue staining, the osteoid appears light blue and the mineralized bone dark blue.

**Table 1 diagnostics-13-01279-t001:** Summary of the main clinical and radiological characteristics for each included patient (case 1–6) with OO (negative (−); positive (+); not known/examined (0)).

Case Number	1	2	3	4	5	6
age/gender	35/m	27/m	22/m	27/m	24/m	29/m
side	left	right	left	left	right	left
bone	prox. phalanx	hamate	capitate	prox. phalanx	dist. phalanx	prox. phalanx
localization	intracortical	intracortical	juxta-articular	juxta-articular	intracortical	intracortical
time to diagnosis (months)	62	7	18	12	12	12
Detection OO in imaging Xray/CT/MRI	−/+/+	−/+/−	+/+/+	−/+/−	−/+/+	+/+/+
widest nidus diameter (mm)	9	6	7	10	6	9
pain/night pain/improvement due to NSAID	+/+/+	+/+/−	+/+/+	+/0/0	+/+/+	+/+/+
swelling/deformity/motoric deficites	+/+/−	−/−/−	−/−/+	+/+/+	+/+/-	+/+/+
treatment	en bloc	curretage	curretage	curretage	curretage	curretage
follow-up (months)	78	4	14	16	16	4

## Data Availability

All data are provided within the manuscript.

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
