# Peer review of "The Diagnostic Challenge of Osteoid Osteoma in the Bones of the Hand—A Case Series"

_diagnostics, 2023, doi:10.3390/diagnostics13071279_

Round 1

Reviewer 1 Report

The article is devoted to an actual topic. The paper describes in detail the clinical data for all clinical cases. However, it is worth adding information about the genetic features of pathology to the discussion

Author Response

Dear colleague,

Thank you for reviewing my manuscript. I implemented the recommended additions. Please read the added paragraph about the possibility of using gene markers to differentiate benign bone tumors such as osteoid osteoma and osteoblastoma from malignant osteosarcoma. You will find the new paragraph in the discussion after the description of the histological features of osteoid osteoma. Feel free to write if you have any further comments.

Reviewer I comment:

The article is devoted to an actual topic. The paper describes in detail the clinical data for all clinical cases. However, it is worth adding information about the genetic features of pathology to the discussion

Added Paragraph:

When comparing the histological presentation of OO with the Osteoblastoma they are histologically indistinguishable [1]. However, the two bone-forming tumors are characterized by the size of their nidus which in Osteoblastoma is defined as more than 2 cm in diameter [2]. Imaging often succeeds in distinguishing OO and Osteoblastoma from the most common primary malignant bone tumor, the Osteosarcoma. This tumor with an extremely high mortality appears as a lytic, sclerotic, or mixed lytic and sclerotic lesion often combined with a reaction of the surrounding soft tissue and destruction of the cortical bone. Histologically, several subtypes can be distinguished [1]. However, recent studies have shown that in case of uncertain differentiation between benign OO or Osteoblastoma and Osteosarcoma, the diagnosis can benefit from using fluorescence in situ hybridization (FISH) analysis for FOS gene arrangement [3]. FOS and at a lower frequency FOSB rearrangements were found in the majority of OO and Osteoblastomas, whereas they have only been detected less frequently in osteosarcomas [1, 3, 4]. This provides an additional tool in the histopathological examination for differentiation of the tumor after surgical resection.

Reviewer 2 Report

The diagnostic challenge of osteoid osteoma in six cases of phalangeal and carpa. The topic is not original. There are many related articles. There are articles that describe dozens and even 100 cases of Osteoid osteoma of the hand and wrist. There is nothing new or interesting in this article and it concerns only a series of 6 patients. To consider a similar article for publication would have to present something new. For example, new diagnostic or therapeutic methods. It should be based on a lot more patients - minimum 20. The conclusions bring nothing new. They are simply written.

Author Response

Dear colleague,

Thank you for reviewing my manuscript. I am sorry that you don’t approve the actual version of the manuscript. Based on recommended changes by the other reviewers I have added some paragraphs about the histological differentiation from other bone tumors. In addition, I have described the possibility of using biomarkers in the case of uncertain histological findings. As recommended by you I added some references. Feel free to write if you have any further comments.

Added Paragraphs to the Discussion:

When comparing the histological presentation of OO with the Osteoblastoma they are histologically indistinguishable [1]. However, the two bone-forming tumors are characterized by the size of their nidus which in Osteoblastoma is defined as more than 2 cm in diameter [2]. Imaging often succeeds in distinguishing OO and Osteoblastoma from the most common primary malignant bone tumor, the Osteosarcoma. This tumor with an extremely high mortality appears as a lytic, sclerotic, or mixed lytic and sclerotic lesion often combined with a reaction of the surrounding soft tissue and destruction of the cortical bone. Histologically, several subtypes can be distinguished [1]. However, recent studies have shown that in case of uncertain differentiation between benign OO or Osteoblastoma and Osteosarcoma, the diagnosis can benefit from using fluorescence in situ hybridization (FISH) analysis for FOS gene arrangement [3]. FOS and at a lower frequency FOSB rearrangements were found in the majority of OO and Osteoblastomas, whereas they have only been detected less frequently in osteosarcomas [1, 3, 4]. This provides an additional tool in the histopathological examination for differentiation of the tumor after surgical resection.

Reviewer 3 Report

This manuscript/case report describes the difficulty in diagnosing osteoid osteoma (OO) in the bones of the hand, by presenting 6 case reports. Although CT and MRI-based imaging is useful, OO is rare in the bones of the hand. This report aimed to highlight diagnostic difficulties and the risk of misdiagnosis. Here are several comments to improve the manuscript.

·       OO cases: OO in the bones of the hand is reported rare. It is recommended to quantitatively present how rare OO is.

·       OO sites:A description regarding the comparison to OO in the bones of other sites can be included in the introduction.

·       Histological diagnosis: In all cases, histological evaluation was conducted to identify OO. The key histological features can be described in this case report. Particularly, the differences between OO and osteosarcoma/osteoblastoma are recommended to be presented.

·       Biofluids and gene markers: The possibility of using biofluids or identifying gene markers can be discussed to face the diagnostic challenge.

·       Scale bar: The scale bar should be included in Figures 2 and 3.

Author Response

Dear colleague,

Thank you for reviewing my manuscript. I made some changes and added some paragraphs based on your comments. I also added the scale bars in Fig. I and II. Feel free to write if you have any further comments.

Reviewer III comments:

OO cases: OO in the bones of the hand is reported rare. It is recommended to quantitatively present how rare OO is.

OO sites: A description regarding the comparison to OO in the bones of other sites can be included in the introduction.

Added Paragraphs to the Instruction:

Osteoid osteoma (OO) was first reported by Jaffe in 1935 [5]. It is a common benign bone tumor mostly present in long tubular bones of the lower extremities. The femur and tibia are the most affected bones, with the OO usually located close the end of the diaphysis. Overall, OO accounts for about 12 % of all benign bone tumors [2, 6].

OO in the bones of the hand is less common and the clinical presentation often non-specific. An evaluation of 367 cases treated at the University Medical Center Hamburg-Eppendorf revealed a frequency of 8.4 % in the bones of the hand [7].

Reviewer III comments:

Histological diagnosis: In all cases, histological evaluation was conducted to identify OO. The key histological features can be described in this case report. Particularly, the differences between OO and osteosarcoma/osteoblastoma are recommended to be presented

Biofluids and gene markers: The possibility of using biofluids or identifying gene markers can be discussed to face the diagnostic challenge.

Added Paragraphs to the Discussion:

When comparing the histological presentation of OO with the Osteoblastoma they are histologically indistinguishable [1]. However, the two bone-forming tumors are characterized by the size of their nidus which in Osteoblastoma is defined as more than 2 cm in diameter [2]. Imaging often succeeds in distinguishing OO and Osteoblastoma from the most common primary malignant bone tumor, the Osteosarcoma. This tumor with an extremely high mortality appears as a lytic, sclerotic, or mixed lytic and sclerotic lesion often combined with a reaction of the surrounding soft tissue and destruction of the cortical bone. Histologically, several subtypes can be distinguished [1]. However, recent studies have shown that in case of uncertain differentiation between benign OO or Osteoblastoma and Osteosarcoma, the diagnosis can benefit from using fluorescence in situ hybridization (FISH) analysis for FOS gene arrangement [3]. FOS and at a lower frequency FOSB rearrangements were found in the majority of OO and Osteoblastomas, whereas they have only been detected less frequently in osteosarcomas [1, 3, 4]. This provides an additional tool in the histopathological examination for differentiation of the tumor after surgical resection.

Round 2

Reviewer 2 Report

The authors corrected the article. Added information about the histological differentiation from other bone tumors. In addition, they added information about the possibility of using biomarkers in the case of uncertain histological findings. Added some references. Acceptable article.